# Psychological safety and patient safety: A systematic and narrative review

**Anthony Montgomery**[1*], **Vilma Chalili**[2], **Olga Lainidi**[3], **Christos Mouratidis**[4], **Ilias Maliousis**[5], **Konstantina Paitaridou**[4], **Alison Leary**[6]

1 Northumbria University Newcastle, Newcastle, United Kingdom, 2 Aristotle University of Thessaloniki, Thessaloniki, Greece, 3 University of Leeds, Leeds, United Kingdom, 4 University of Derby, Derby, United Kingdom, 5 University of Essex, Essex, United Kingdom, 6 London South Bank University, London, United Kingdom

* anthony.montgomery@northumbria.ac.uk

## Abstract

### Objectives

Various psychological concepts have been proposed over time as potential solutions to improving patient safety and quality of care. Psychological safety has been identified as a crucial mechanism of learning and development, and one that can facilitate optimal patient safety in healthcare. We investigated the quantitative evidence on the relationship between psychological safety and objective patient safety outcomes.

### Methods

We searched 8 databases and conducted manual scoping to identify peer reviewed quantitative studies published up to February 2024. Objective patient safety outcomes of any type were eligible. The findings were analysed descriptively and discussed in a narrative synthesis.

### Results

Nine papers were selected for inclusion which reported on heterogeneous patient safety outcomes. Five studies showed a significant relationship between psychological safety and patient safety outcomes (e.g., ventilator associated events, reported medical errors). The majority of studies reported on the experiences of nurses working in healthcare from the USA. Patient safety is consistently characterised as the absence of harm rather than a culture that creates a safe environment.

### Conclusions

No clear conclusions can be extracted regarding the relationship between psychological safety and patient safety. For example, reporting patient safety problems in a team can be an indication of both high and low psychological safety. Patient safety

**Data availability statement:** All relevant data are within the manuscript and its Supporting Information files.

**Funding:** The author(s) received no specific funding for this work.;

**Competing interests:** The authors have declared that no competing interests exist.

may be contradictory to elements of psychological safety, as the absence of harm is not congruent with a safety environment approach. Systematic review registration: This systematic review is registered with the International Prospective Register of Systematic Reviews (PROSPERO CRD4202347829).

## Introduction

Psychological safety (PS), which describes a work environment where people believe that candor is expected and possible, is particularly important for patient safety. PS has matured as a core concept in management and psychology, as the number and reach of studies have grown exponentially [1–2]. While PS has been linked to lower job burnout levels [3] and a supportive work context [4], Edmondson and colleagues [2] pinpoint the limitations, which include a significant lack of knowledge about creating PS and the role other team members play in this, measurement in non-Western countries, and lack of longitudinal research. In healthcare, a recent research synthesis found that substantial variations in PS were reported by healthcare workers across all studies, and evidence that there is an ongoing need to focus upon its improvement [5]. Moreover, there is evidence that PS doesn't work in the expected way in healthcare [6–8], where figuring out what can be talked about (and not talked about) trumps the generic idea of PS per se. This results in a paradoxical phenomenon, whereby high PS teams appear to operate less safely due to their increased tendency to report more unsafe practices and the willingness to report incidents [9,10]; such facts raise questions about the role and function of PS within medical practice suggesting that it represents an organizational culture element rather than a straightforward determinant of patient safety culture. PS may seem to operate differently within healthcare, stressing the reconstruction of error management culture, where errors are viewed as ongoing learning and improvement in safety practices, fostering collective accountability, and mutual trust rather than the straightforward equation of the absence of reported errors with increased patient safety indicators. This perspective may challenge conventional organizational norms that equate fewer errors and mistakes with success [11]. To better understand why this is happening it is necessary to evaluate the existing evidence on the relationship between PS and patient safety outcomes, providing a deeper analysis that can delineate what elements of PS are linked with patient safety.

### Psychological safety and patient safety

The relevance of PS to medical and clinical settings lies in its facilitation of intuitive processes/reasoning [12] such as fostering open team communication, trust, interpersonal evaluations, and goal-directed behaviour (e.g., intention to benefit patients and avoid harm) that can enhance social team interactions by removing barriers of reprisal and fear [13]. Traditionally, PS has been linked to positive outcomes that include learning [14–15], creativity and proactivity [16], innovation [17], improved practice [18] and adaptation to changes [19]. Despite these more positive outcomes,

there is emerging evidence that 'too much' PS has negative consequences [6,20], meaning that it is hard to know what psychological safety is NOT. For example, Eldor et al [6] in their research on professional actions among nurses found that when tasks are routine, high levels of PS climate can harm in-role performance potentially due to cognitive distraction or task experimentation. Interestingly, Jung et al [21] while exploring the relationship of PS and objective professional actions (e.g., the use of an incident and near-miss reporting system) from various professional groups involved in delivering care for radiation oncology found an ever more complex outcome. PS did not correspond to the use of the reporting system as professions with lower PS were more likely to use the reporting system compared to physicians who expressed the highest PS. A critical consideration regarding the challenges of in-role performance and identity on the commitment to learning from incidents and errors arises from this perspective. It can be also assumed that when systems fail to provide meaningful feedback or translate incident reports into actionable improvements, PS "loses" undermining its role in promoting a culture of continuous improvement and reciprocal communication. As a result, when talking about PS, more attention may have been given to healthcare professionals' willingness to report errors and, thus neglect its impact on actual objective actions of reporting – while both subjective intentions and objective actions balance medical accountability [22]. In fact, it is well-established across decades of research on the links between intentions and behaviours that self-reported intentions do not always align with objective observable actions in clinical settings. However, as intentions are usually measured in a vacuum with generic questions (e.g., not linked to specific context), it should not be surprising that such theoretical predictions do not hold in practice. For example, while an increased amount of literature has subjectively measured reasons why healthcare workers do not speak up, so far interventions attempting to increase voice based on that evidence are not succeeding [23]. Within healthcare, safety culture highlights the importance of how subjective perceptions and beliefs could positively influence attitudes and objective actions related to safety [24,25]. However, considering the criterion of hospital safety – despite the well-meaning intentions – a plethora of error reporting rates may obscure critical safety signals in the health industry [26]. This emphasis on quantity (fewer errors reported) over quality (leveraging reported errors for ongoing learning, targeted analysis, and improvement) [27] conflicts with the core principles of PS within hospital safety frameworks. Defining the commitment to advancing safe care, patient safety—the most enduring and foundational principle of medicine—represents the core value of healthcare quality by emphasizing freedom from any harm associated with health care in clinical practices [28]. "*An organisation with a memory*" established the perspective beyond individual accountability recognising that errors in medication administration concerns an even more intricate and complex causation [29]. The field of patient safety has been informed by the area of safety compliance, which refers to following safety protocols meant to protect both employees and patients. Following protocols is associated with fewer occupational injuries [30–31], however PS might mean that justifiable deviance is warranted, as recommended by the Safety II approach [32]. Medical errors translate into over three million deaths globally each year [33] and are a key contributor to provider depression, posttraumatic stress disorder, suicidality, impaired work performance, burnout, and turnover [34]. Sources of medical errors include the actions of health care professionals, safe care system failures (e.g., communication failures), or a combination of errors made by individuals, system failures, and patient characteristics [35].

PS, with its focus on high-quality communication, trust and decision-making is assumed to play an important role within workplace teams generally, and particularly in healthcare, with the notion that when healthcare teams are psychologically safe, they are more likely to engage in quality improvement and team learning initiatives [36–37]. Moreover, effective working relationships have been tied to improved quality of patient care in a variety of clinical settings [38–40]. According to the upward voice communication framework, in psychologically safe work environments, employees—regardless of their identity role (e.g., nurse, physician), seniority, or frontline status, whether first-line staff, middle management, or executives — shouldn't hesitate to report safety concerns regardless of power differentials as speaking up should be seen as a contribution to safety rather than wrongdoing or personal failure [41]. However, regardless of the upward communication framework, a historical stigma is attached to error reporting in healthcare, as in highly complicated environments, this could lead to personal accountability [42]. The essential problem is that increasing PS in a team can lead to negative and

opposite effects on positive risk-taking behaviours via fear of failure and decreased work motivation [43]. Therefore, the direction or existence of causality between patient safety and PS is not clear.

### Study objectives

This systematic narrative review aims to summarize and clarify the existing evidence concerning PS and patient safety outcomes in healthcare to uncover the conceptual, theoretical and methodological challenges in linking the two. The review included only studies that used robust measures of patient safety outcomes linked to observable professional actions or/and reporting behaviours as opposed to self-report ratings of perceived patient safety, given the evidence that self-reports can artificially inflate the relationship between safety climate and safety outcomes [44].

The following research question was considered: What evidence supports the relationship between PS and actual patient safety outcomes? There is a bigger question to consider as to what extent increased reports of patient safety incidents and unsafe practices indicate a culture of transparency and learning rather than a true decline in safety performance. The review aimed to gain insights for interventions and policy in health and social care, aligning the concepts of learning and clinical performance to foster innovation in error management towards a high-quality multiperspective approach to safety culture.

## Methods

The high heterogeneity of outcomes did not allow for meta-analytic synthesis; thus, a narrative approach was deemed most appropriate.

### Search strategy

This review was developed following the PRISMA guidelines [45] and the accompanying PRISMA Checklist is available in S1 File.

This systematic review included only studies with robust quantitative indicators as they employ systematic and epidemiological methods to quantify specific aspects of objective patient safety measures, including the use of the reporting system and checklists, adverse events, harm to patients and providers, and risks associated with adverse events [46]. Time and country restrictions were not applied. The electronic databases consulted with language restrictions to English were PubMed, PsycINFO, Scopus, Embase, Cochrane Library, Web of Science, CINAHL and the search was supplemented by Google Scholar and manual scoping (S2 File). The review searched databases up until February 2024.

**Eligibility criteria and study selection.** The exclusion/inclusion criteria are available in S3 File. Duplicate control and title and abstract review were conducted using Rayyan. Two authors independently first screened for inclusion by title & abstract (n = 573) and then conducted a full-text review (PRISMA flow diagram Fig 1, reasons for exclusion see S4 File). Cohen's kappa indicated substantial inter-rater agreement for title & abstract screening (k = 0.95, 98.2%). Regular meetings with a third reviewer allowed discussion of article eligibility resulting in a total of eighty-nine articles for full-text screening. Low agreement (k = 0.36, 79%) was found for the full- text review due to the vastly heterogeneous outcomes related to patient safety measures (e.g., perceived level of patient safety versus metrics of patient-safety outcomes). A third reviewer reviewed all the 89 articles independently and a fourth reviewer examined conflicting decisions. Addressing the above-mentioned conflicts resulted in the final inclusion of nine articles for review [47–55].

**Quality assessment and data extraction.** Two authors independently extracted all the collected data (see S5 File for extracted data) and conducted quality assessments for included studies using the Quality Assessment Tool for Observational Cohort and Cross-Sectional Studies [56]. Quality assessment was also independently reviewed by other two authors (k = 0.81, 90.5% almost perfect agreement;). Eight studies were rated as Fair (score > 7 < 11) and one as Good (for details per study see S6 File).

## Figure 1. PRISMA Flow Diagram

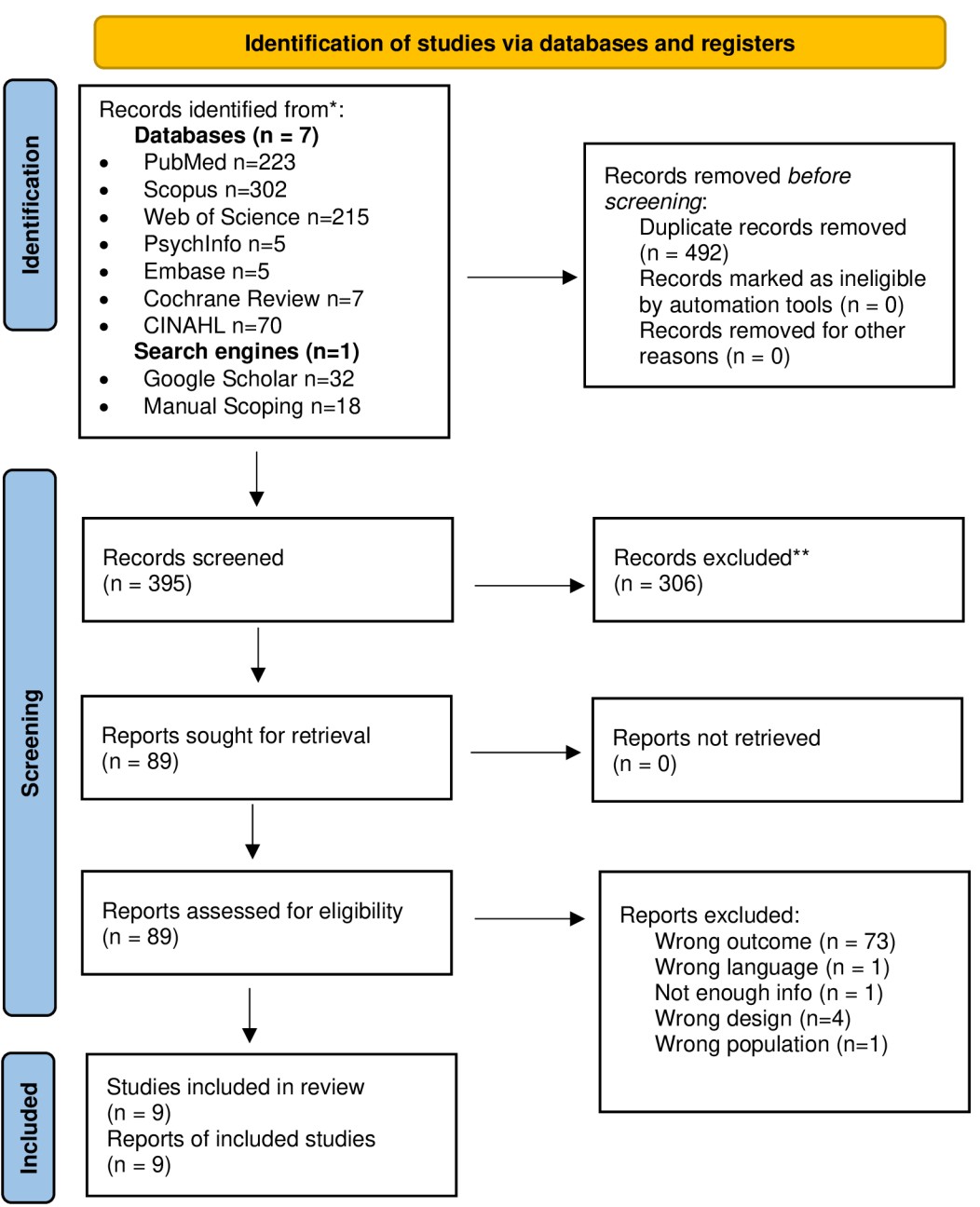

**Fig 1. Prisma flow diagram.**

## Results

### Descriptive information on the studies

The total sample of participants was N = 17,926. Gender representation was noted in only 44% (4 out of 9) of the studies. Of the total sample, 88.4% were nurses. Detailed information on study characteristics is available in Table 1. Almost all

included studies (8/9) were conducted in the USA and most were published between 2017–2023. Six of the studies used Edmondson's PS questionnaires. Two studies used items from the Workgroup psychological safety questionnaire of the Veterans Health Administration's "All Employee Survey". One study used items from the Learning Organization Survey (LOS). The following patient safety outcomes were limited to objectively observable professional actions and reporting behaviours (i.e., reporting of medical errors, treatment errors, medication errors, incidents, near misses, occupational injuries, following protocol/checklist, the use of seclusion and physical restraining).

**Is there a relationship between psychological safety and patient safety?.** PS and positive objective patient safety outcomes are sometimes "difficult" to detect and resistant to linear interpretation, as PS seems to share contextual dependencies between objective reporting behaviour and patient safety. Additionally, high reporting rates may indicate high PS, where incidents are openly discussed and used as learning opportunities and protocol compliance. Conversely, low reporting rates might be interpreted as evidence of enhanced team performance, although this could also reflect underreporting or fear of disclosure. Of the nine studies, only five studies indicated a significant relationship between PS and patient safety [50–52,54,55]. However, all five studies need further qualification to fully appreciate the complexity in linking PS and patient safety. The influence of PS on patient safety varies significantly depending on the specific outcomes measured, the context in which safety incidents/error occur, and the kind of incidents/error. Anderson et al, [51] in a sample of inpatient psychiatry clinicians, found that units with lower PS reported a greater use of restraints with patients, but also found that units with higher PS utilized seclusion more. This outcome is complex and somewhat paradoxical, as seclusion is considered a last-resort strategy to ensure safety [57–58]. According to the results of the study feelings of PS—or the lack thereof—influence healthcare providers' direct behaviour and, consequently, the *kind of care* (seclusion vs physical restraints) provided to patients.

Halbesleben et al,[50] in a sample of nurses, found that leader behavioural integrity and the reporting of medical errors was related when PS was high. These findings suggest that the degree to which employees feel psychologically safe directly impacts their willingness and ability to comply with safety protocols. Anderson et al, [51] and Halbesleben et al, [50] findings may indicate behavioural dynamics underlying PS and patient safety complex trade-offs. Brimhall et al, [55] in a sample of 318 healthcare employees, found a negative association between PS and reported medical errors, suggesting that high-PS teams may have open dialogue among team members about errors to learn from mistakes and improve systems to reduce future errors. Also, the idea that high-PS doesn't lead to progress in the short term was reflected in Leroy et al, [52] who also found that PS had a negative relationship with the number of reported treatment errors, with the authors speculating that this relationship could be reversed in the long run. PS alone might increase the reporting of errors, which initially has an opposite effect on the safety records. However, when cultivating both a strong adherence to safety protocols (priority of safety) and a supportive error management where staff feel secure in voicing concerns and reporting errors (PS) it could point to a safety culture where errors are not *just* reported but also decreases over time.

Jung et al, [54] in a sample of radiation oncology department staff, presented five scenarios about a patient with a cardiac pacemaker (arranged in order of proximity to failure) and showed that PS is an important predictor of reporting incidents and selected near-miss types. However, the effect of PS on reporting near-miss events became stronger only with the events' increasing proximity to a negative outcome. These results reflect the question of how safety protocols are placed regarding reporting incidents such as "actual harm" and "almost happened", as it is indicated that PS appears to operate differently depending on the kind/type of incident. Overall, the evidence linking psychological and patient safety is equivocal.

In the remaining four studies, Arnetz et al, [47] in a sample of nurses, found that Unit-level PS was not significantly associated with Central Line-associated bloodstream infections, Catheter-associated urinary tract infections or Ventilator-associated events. Ridley et al, [48] examining a teamwork training program among cardiothoracic operating room members, found that while there was a trend towards lower error rates, the rates of positive PS measured at the group level did not change significantly during their study. Gilmartin et al, [49] in a sample of nurses, found no statistically significant

**Table 1. Study Characteristics (n = 9).**

| Authors (Alphabetical) | Country | Participants | (Total N, % Women) | Design | Measure of Psychological Safety | Patient Safety Outcome | Study Quality[b] |
|---|---|---|---|---|---|---|---|
| Anderson et al. (2021) | United States | Mental health (n = 797) Registered Nurses (n = 4331) Licensed Practical Nurses (n = 1518) | 6646, NR | Cross-sectional Retrospective Database Analysis | Workgroup psychological safety from the "All Employee Survey" within the Veterans Health Administration | The use of seclusion and physical restraining in inpatient psychiatric units. | 9/14 |
| Arnetz et al. (2019) | United States | Nurses in the hospital | 432/95.1% 83 blood samples; 95.1% | Cross-sectional Retrospective Analysis | Edmondson (1999) | (1) pressure ulcers (2) patient falls (3) central line-associated blood stream infections (CLABSI) (4) catheter-associated urinary tract infections (CAUTI) (5) ventilator-associated events (VAE) - All unit level | 7/14 |
| Brimhall et al. (2023) | United States | All employees of a non-profit hospital from various departments | 318 employees from 47 workgroups; NR | Cross-sectional | Edmondson (1999) | Reported medical errors | 8/14 |
| Gilmartin et al. (2018) | United States | Nurses working in a Veterans Health Administration hospital | 2008 1,962; 78.19% 2009 1,926; 77.21% 2010 2,428; 76.89% 2011 1,973; 75.98% | Cohort Study - Retrospective Database Analysis | One item from "All Employee Survey" | Nonadherence rates to the central line checklist: (1) hand hygiene before central line insertion (2) application of chlorhexidine gluconate (prep) (3) use of a cap (4) mask, (5) sterile gloves, (6) sterile gown by the provider inserting the central line (7) full-body drape to cover the patient | 7/14 |
| Halbesleben et al. (2013) | United States | Registered Nurses | 658, 87% | Cross-lagged study | Edmondson's (1999) adapted version (Nembhard & Edmondson, 2006) | Occupational Injuries | 9/14 |
| Jung et al. (2021) | United States | Staff of a Radiation Oncology Department | 78, NR | Cross-sectional | Learning Organization Survey (LOS) | Willingness to report incidents (near misses and therapeutic incidents) | 9/14 |
| Leroy et al. (2012) | Belgium | Nurses and Head nurses from various specialty departments | nurses = 580; 75% head nurses = 54; 56% | Cross-lagged | Safety For Nurses: Simons et al. (2007). PS For Teams: Edmondson (1999) | Reported Treatment Errors that resulted in harm to a patient. | 9/14 |
| Raman & Green, (2017) | United States | Non-physician healthcare professionals | 803, NR | Cross-sectional | Edmondson (1999) | Medication administration processes in healthcare settings/ records | 9/14 |
| Ridley et al. (2020) | United States | Operating Rooms (ORs) Clinicians | 73 at Baseline 6-month Follow-Up: 68 12-month Follow-Up: 68 NR | Cohort Study | Edmondson (1999) | Medical Errors Reported During Surgical Cases (defined as a preventable adverse event resulted OR NOT in harm to a patient) | 11/14 |

a Data Extraction for included articles.

b Quality assessment is performed with the use of the Quality Assessment Tool for Observational Cohort and Cross-Sectional Studies (Feng et al., 2014). The tool contains 14 criteria, and the evaluator is asked to answer whether the study in question meets the criterion. The possible answers are Yes, No, Cannot determine, Not applicable, and Not reported. A score of > 11 corresponds to good quality, 7–10 to fair quality and < 7 to poor quality.The breakdown of individual scores can be found in S5 File.

*NR=Not Reported

differences in PS scores for units with 5% or less checklist nonadherence and units with more than 5% checklist non-adherence within any year of the data. Raman & Green, [53] in a sample of non-physician clinician workforce across 27 patient-care units, the relationship between employees' perceived PS and their proportion of timely medication use was non-significant.

**Methodology and sampling issues.** The majority of studies used the Edmondson questionnaire to assess PS (7/9). The use of a common metric is desirable and provides the opportunity for comparison. The Edmondson questionnaire prompts individuals to report on what is permissible in their team/workgroup. However, this begs the question as to what team do individuals have in mind when they fill in these questionnaires. We don't know and this may explain the ambiguous relationship between PS and patient safety. As noted by Edmondson & Bransby [2], the literature on PS thus far provides relatively little insight on how interpersonal climates change over time and on how or when to intervene productively. Other unaccounted for factors may also influence the way that PS is understood by different members of the team. For example, Grailey et al, [5] in an evidence synthesis of PS in healthcare, concluded that many of the factors that contribute to PS are not malleable or easy to change (especially within a resource poor environment). Thus, knowing *why* PS is low may be as important as knowing if it is low. Meta-analytic evidence indicates that internal consistency can be low for PS measures (i.e., alpha <.70 or <.79) [59]. This may reflect the fact that not all elements of PS are necessarily present in the same team/organization or not all elements are experienced and perceived the same by all team members/employees, with implications for whether treating PS as a latent variables is wise or approrpaite

The representativeness of the nine studies reviewed was limited. The majority of the studies (8/9) were conducted in the US and reported on the experiences of nurses and women. Thus, even for the five studies that reported an association, our ability to generalise is extremely restricted, and review findings might be more representative of the research and experiences of nursing professionals and of women working in healthcare. To achieve a better understanding of the complex phenomena of PS and patient safety in healthcare, future research needs to address all involved professional groups. Additionally, the US centric character of the reviewed papers means that it not only suffers from the so-called "WEIRD" problem—meaning that the majority of research is usually carried out in western and developed countries – but we also can't be sure whether the findings are generalizable outside the US.

**Direction, timing and causality.** The review highlighted some contradictions about the timing and mechanisms of PS. Certain papers suggested that higher levels of PS should be associated with more error reporting (i.e., Leroy et al [52]), while others suggested the opposite (i.e., Anderson et al, [51]). The former was linked to the notion that increased feelings of PS should lead to more reporting, whereas the latter is linked to the idea that less error occurrence will be a consequence of a more "healthy" team/unit environment. The crucial issue is the development of PS over time – whether it's at the beginning of the process or well established within the team. For example, in the study of Ridely et al [48] the rates of positive PS measured at the group level did not change significantly during their study, but satisfaction with teamwork and feeling comfortable to speak up measured on a daily basis did show improvement over the period of the training. Thus, PS appears to work differently comparing group and daily measures. Moreover, while the number of medical errors decreased, error reporting did not significantly increase, leading the authors to speculate that individuals no longer felt that it was helpful to report minor errors to the hospital system due to better communication between employees. It's not clear what the causal mechanism linking feelings of safety and reporting is. The study of Anderson et al, [51] with inpatient psychiatry clinicians, found that units with lower PS made more use of physical restraint among patients, but those with higher PS reported greater use of seclusion strategies. Significantly, PS is considered a cognitive concept [60] thereby linked to emotional states that in healthcare providers could lead to biases in decision-making, impacting patient care practices related to safety and quality [61]. The authors attempted to explain these contradictory findings by reflecting on whether seclusion use is viewed as relatively noncoercive or a less coercive way of managing violent/disruptive behaviour. Thus, it could be the "lesser of two evils". In the Leroy et al, [52] study of nurses, the authors suggest that higher levels of PS could result in the reverse results over time (i.e., less reported errors) because an

environment supportive of reporting errors can help employees learn from mistakes in the long run. Jung et al, [54] in their scenario study, speculate that improving PS may not increase recognition of near misses, particularly those that more closely resemble standard care than an incident. The influence of PS on reporting is more pronounced in scenarios where near misses are perceived as more critical or dangerous.

In the Arnetz et al, [47] study, PS was associated with self-reported stress and competence development but not significantly with biological markers or objective patient outcomes. The Gilmartin et al, [49] study concludes that their data neither supported their hypothesis or previous research findings concerning the link between PS and error reporting. The reflection by the authors on the results is revealing of the confusion surrounding PS. Potential reasons for the lack of association include the presence of the observer producing an improvement in performance (i.e., Hawthorne Effect), reporting on nonadherence being perceived as risky and a fear that they may be personally blamed for not taking ownership which resulted in many nurse respondents selecting the neutral response, "neither satisfied nor dissatisfied" on the PS question. Overall, the discussion sections of the reviewed papers provided interesting speculations regarding the relationship between PS and patient safety. However, it's not a healthy sign for the field that diverse speculations are so numerous.

## Discussion

Overall, there is relatively little hard data to link PS and patient safety outcomes. Only nine studies fit the criteria that examined PS and objective measures of patient safety. This is stark contrast to literature that purports a clear link between the two phenomena [5,62,63]. The findings of the review imply a contradiction in patient safety practices: enhancing team dynamics through PS culture may improve immediate problem-solving within the team, but it does not automatically translate into improved objective patient safety measures.

### Potential reasons for the lack of evidence

The simplest and initial point to accept is that we simply don't have enough research yet to establish a link between PS and objective measures of patient safety. Absence of evidence is not evidence of absence. However, that caveat should not prevent us from discussing the potential factors influencing the relationship. For example, a line manager may espouse the importance of safety procedures while they fail to enact, enforce, and support the same safety procedures through their actions via monitoring and allocation of time and resources. As a result, employees may experience a double bind between these seemingly conflicting behaviours [64] (p. 117): "…when employees adhere to a norm that says, "hide errors," they know they are violating another norm that says, "reveal errors""The employees are thus in a double bind.

As noted by Halbesleben et al, [63] looking at only one indicator (e.g., frequency) may not represent the whole picture of safety, whereby a low frequency of injuries may actually be an indication of low reporting rather than an indication that the organization scores high on safety. Congruently, severe injuries are usually more heavily controlled by protocols, meaning 'less serious' problems are less likely to be reported. Thus, we may need to examine which type of patient safety outcomes link with PS. Additionally, the timing of the patient safety events is critical. As noted by Hirak et al, [65] near misses that occur early in the process of care may be perceived as cognitively distal to the averted failure, thus underscoring resilience. In contrast, near misses that occur later in the process may be perceived as cognitively proximate to the averted failure, thus underscoring vulnerability. This distinction between what represents resilience and vulnerability is especially pertinent in the healthcare industry where risk is a constant concern. However, the definitions of patient safety in the literature reviewed, largely focus on the prevention of harm rather than creating a safety culture in which PS is an integral part.As noted by Reason, safety is more than the absence of harm [66].

Limitations concerning the present review centre around the fact that patient safety metrics and medical accountability represent concrete phenomena, while PS is an abstract psychological phenomenon. However as stated above, this narrow view of safety as absence of harm could be broadened. The current literature focuses on what can be measured

in terms of healthcare performance (harm) and not what is perceived (safety). What can be measured then becomes the focus, and increasingly a performance target. When a measure becomes a target, it ceases to be a good measure and the change in focus may account for the discrepancy in findings [67]. Limitations on definitions and operationalizations also concern patient safety, given that patient safety outcomes are not universal and even basic definitions differ across countries – especially terms like "medical errors", which concurrently function as legal terms [68]. According to Kaldjian et al, [69] even in teaching hospitals, a gap exists between the intention to report and the actual act of reporting medical errors, due to severe repercussions; thus, there might be scope for PS to be more important in predicting the expected/ possible errors rather than the reported/occurred ones. Some common reasons that may lead to underreporting behaviour include fear of legal complications, fear of peer judgement/disapproval, negative attitudes toward reporting errors, lack of time, and the complex process of reporting [70–73]. Nurses are often the most frequent reporters of incidents, compared to physicians [74], which means a more heterogeneous sample may produce different outcomes.

## Conclusions

Ultimately, we are left with a paradox regarding PS in healthcare teams. Reporting patient safety problems in a team can be both an indication of high and low levels of PS. It's difficult to know which without understanding the culture and history of the specific healthcare organization, as PS primarily impacts emotions and attitudes rather than patient safety metrics directly. The most reliable evidence concerning the benefits of PS relate to creative/learning activities, however it can be detrimental concerning routine tasks. The way that PS is assessed needs further exploration, as it's not yet clear what individuals have in mind when reporting on the climate in their team for sharing information. Psychological and patient safety may not be easily aligned. The paradox between PS and patient safety lies in their contrasting definitions and goals. Patient safety aims to prevent harm in clinical settings, but this view may be too narrow to assert a relationship with PS in healthcare organisations. In contrast, PS promotes a culture of interpersonal risk-taking within teams. This inherent conflict arises because risk-taking, fundamental to PS, contradicts the principles of patient safety grounded in established medical protocols.

## Supporting information

**S1 File  PRISMA checklist.**
(DOCX)

**S2 File.  Search strings.**
(DOCX)

**S3 File.  Inclusion-exclusion criteria.**
(DOCX)

**S4 File.  Reasons for exclusion.**
(PDF)

**S5 File.  Extracted data.**
(PDF)

**S6 File.  Quality assessment.**
(PDF)

## Acknowledgements

No acknowledgements.

## Author contributions

**Conceptualization:** Anthony Montgomery, Olga Lainidi.

**Data curation:** Vilma Chalili, Christos Mouratidis, Ilias Maliousis, Konstantina Paitaridou.

**Formal analysis:** Anthony Montgomery, Vilma Chalili, Christos Mouratidis.

**Methodology:** Anthony Montgomery, Olga Lainidi.

**Project administration:** Anthony Montgomery, Olga Lainidi.

**Supervision:** Anthony Montgomery, Olga Lainidi.

**Writing – original draft:** Anthony Montgomery, Vilma Chalili, Olga Lainidi.

**Writing – review & editing:** Anthony Montgomery, Vilma Chalili, Olga Lainidi, Christos Mouratidis, Ilias Maliousis, Alison Leary.

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
