## [Decision Letter · Decision Letter 0]

18 Dec 2024

PONE-D-24-42345Psychological safety and patient safety: a systematic and narrative reviewPLOS ONE

Dear Dr. Montgomery,

Thank you for submitting your manuscript to PLOS ONE. After careful consideration, we feel that it has merit but does not fully meet PLOS ONE’s publication criteria as it currently stands. Therefore, we invite you to submit a revised version of the manuscript that addresses the points raised during the review process.

We look forward to receiving your revised manuscript.

Kind regards,

Anna Rachel Conolly, PhD, MSc, PG Dip, BA (hons)

Academic Editor

PLOS ONE

Journal Requirements: When submitting your revision, we need you to address these additional requirements. 1. Please ensure that your manuscript meets PLOS ONE's style requirements, including those for file naming. The PLOS ONE style templates can be found at https://journals.plos.org/plosone/s/file?id=wjVg/PLOSOne_formatting_sample_main_body.pdf and https://journals.plos.org/plosone/s/file?id=ba62/PLOSOne_formatting_sample_title_authors_affiliations.pdf 2. As required by our policy on Data Availability, please ensure your manuscript or supplementary information includes the following:  A numbered table of all studies identified in the literature search, including those that were excluded from the analyses.   For every excluded study, the table should list the reason(s) for exclusion.   If any of the included studies are unpublished, include a link (URL) to the primary source or detailed information about how the content can be accessed.  A table of all data extracted from the primary research sources for the systematic review and/or meta-analysis. The table must include the following information for each study:  Name of data extractors and date of data extraction  Confirmation that the study was eligible to be included in the review.   All data extracted from each study for the reported systematic review and/or meta-analysis that would be needed to replicate your analyses.  If data or supporting information were obtained from another source (e.g. correspondence with the author of the original research article), please provide the source of data and dates on which the data/information were obtained by your research group.  If applicable for your analysis, a table showing the completed risk of bias and quality/certainty assessments for each study or outcome.  Please ensure this is provided for each domain or parameter assessed. For example, if you used the Cochrane risk-of-bias tool for randomized trials, provide answers to each of the signalling questions for each study. If you used GRADE to assess certainty of evidence, provide judgements about each of the quality of evidence factor. This should be provided for each outcome.   An explanation of how missing data were handled.  This information can be included in the main text, supplementary information, or relevant data repository. Please note that providing these underlying data is a requirement for publication in this journal, and if these data are not provided your manuscript might be rejected.

**Additional Editor Comments:**

Dear Authors,

Three thorough reviews of your work have be returned. I find reviewer the third reviewer's comments to be particularly helpful - i.e. reframe the research question in a manner which recognises the complexity of the link between high psych safety and the appearance of a greater number of patient safety incidents and unsafe practices. This could then be unpicked in great detail in the results section. I look forward to reading the revised paper.

Reviewers' comments:

Reviewer's Responses to Questions

**Comments to the Author**

1. Is the manuscript technically sound, and do the data support the conclusions?

Reviewer #1: Yes

Reviewer #2: Yes

Reviewer #3: Yes

2. Has the statistical analysis been performed appropriately and rigorously?

Reviewer #1: N/A

Reviewer #2: N/A

Reviewer #3: N/A

3. Have the authors made all data underlying the findings in their manuscript fully available?

Reviewer #1: No

Reviewer #2: Yes

Reviewer #3: Yes

4. Is the manuscript presented in an intelligible fashion and written in standard English?

Reviewer #1: Yes

Reviewer #2: Yes

Reviewer #3: Yes

5. Review Comments to the Author

Reviewer #1: Introduction:

- Major: The problem analysis does not fit the research question. Bade on the introduction, the reader expects the paper to focus on the complex relationship between PS and outcomes and especially the underlying mechanisms. In stead the paper focuses on the evidence of the relationship and its implications for interventions. Following the introduction, the research question should be wider and especially include the subjective patient safety outcomes in order to unravel this complex relationship. OR the introduction should be complete rewritten in order to focus on why we need a summary of evidence for this specific relationship, why is it so important to show the reader this overview of evidence. In addition, also argue why this overview does not include the subjective safety outcomes.

- Minor: Why is there a heading called “psychological safety and psychological safety”

Methods:

- In the future you might consider using tools to assist you in a systematic review, such as Ryyan or ASreview

- You might consider identifying other reviews on PS. What did and didn't they conclude. Did they also include your selection of papers. Have you identified new papers or was this review also be able to be conducted based on the present reviews?

Results:

- In the interpretation of the relationship there is a lack of nuanced. If a relationship between PS and medical errors is found, this means that people are more or less willing to report errors, it is not the same as the actual errors made. One could argue both ways: high PS could make the reporting of errors less urging and therefore lower OR high PS could make the reporting of errors more natural in an open and learning environment. This is somewhat later on discussed but not incorporated in the presentation of the results at the beginning of the Result section.

- There is a lack of insights presented in teamwork in healthcare. People that filled in the questionnaire of Edmondons are healthcare professionals. They are part of multiple teams; multi teammembership. What are the teams we are referring to in the result section. Are that the multidisciplinairy teams that provide care? Or the monodisciplinairy teams? That is a big difference in relation to PS. In multidisciplinairy teams the different disciplines are stables but not the people representing this discipline.

Discussion/conclusion:

- What is the answer to the second part of the research question?

- Not clear what this review adds. The paradoxes where already know prior to the research. Make clear what we did not know and after the review do know.

- After rewriting the introduction, have in mind that the introduction and discussion should cover other angles. Now they are too similar in their message.

Reviewer #2: This is a really interesting and important paper and congratulations to the authors. The comments made here are in the spirit of strengthening the paper.

The definition of psychological safety is important. Edmondson’s 1999 definition and measurement is not just about me being able to take risks (express vulnerability) in relation to what I do and do not know but also relates to me being able to question my colleagues on their actions and how correct/safe they are. E.g. it is an important point made in the introduction that in healthcare a lot of time is spent trying to figure out what is safe to talk about – but is this not what psychological safety is about?

There are more nuanced definitions of patient safety other than freedom from harm and as you argue this is not a good definition. I think these other definitions informed by Safety Management Systems work in aviation and other fields should also be included here?

Section – Psychological safety and should be ‘patient’ safety?

Line 80 - Medical errors third leading cause of death – needs to be nuanced

https://qualitysafety.bmj.com/content/26/5/423

Lines 85-90 Should psychological safety not also lead to increased reporting of adverse events and to increased challenges to each other on unsafe practices?

Methods

Search strategy

Line 112 – why were only quantitative studies included – needs justification

Results

Line 138 not sure why female participants pulled out in particular – would it be more appropriate to say ‘Only 4 out of 9 studies noted gender of participants’.

Line 142 needs more discussion on what tools other than Edmondson were used.

Discussion

Line 279 – or improved objective patient safety measures (rather than just formal reporting or errors?) and what are these?

Reviewer #3: This an important systematic review of the evidence linking psychological safety and patient safety. The results will be of great interest to specialists in the patient safety domain. I think the manuscript could be greatly improved if the authors were to present their review from a different stance. Rather than presenting a list of the findings of the nine studies, I suggest a more nuanced approach which recognises the complexity of the link between the two entities. We know already that healthcare teams with high psych safety can appear to be operating less safely because they tend to report more patient safety incidents and unsafe practices. This could be the main focus of the research rather than a straight forward yes/no research question. The sections towards the end of the manuscript nicely address this and they could be moved to the beginning of the manuscript as the background to the review.

I think the results section should tell a 'story' rather than a 'a laundry list' of study findings. The study findings could be used as evidence for the 'story'. This would address an issue in the content where the authors report the study findings and then make a conclusion. It would be more interesting to reverse this process.

In my view (a) and (b) in the research question are not necessary and take from the impact of the question itself.

I also have some typos and writing style issues spotted which I hope are helpful:

The 0-10 rule is inconsistently applied. i.e. numbers between 0-10 should be written in long hand (abstract and L139, 8/9 7/9 etc )

L57, Should 'of' be deleted ?

Ideally sentences that begin with XXXXX found/demonstrated ........are not ideal....Instead state the finding and include the author reference at the end where possible

L 151 inconsistent use of capital letters

L92 rewrite' Therefore the direction or existence of causality between patient and PS is not clear'

Psychological safety is shortened early on to PS but this is not then used consistently ie Lines 96,207,209,214,228,255,256,259,262,264,268,270,273,274,277,282,296,304,307,318,325,330,331,334,337,336,337,338. Personally I prefer the phrase rather than the acronym as its easier for the reader.

L 225 paper or papers ?

I would love to see this manuscript published as it would add value to the body of knowledge if it was to be rewritten more creatively.

6. PLOS authors have the option to publish the peer review history of their article (what does this mean? ). If published, this will include your full peer review and any attached files.

**Do you want your identity to be public for this peer review?** For information about this choice, including consent withdrawal, please see our Privacy Policy .

Reviewer #1: No

Reviewer #2: No

Reviewer #3: **Yes: ** Eva Doherty

---

## [Author Response · Author response to Decision Letter 1]

23 Jan 2025

Response to reviewers

Dear Dr. Connolly and Reviewers,

We want to thank you and the three reviewers for their constructive feedback on our paper. Their suggestions improved the paper. In the following letter, we address all three reviewers' comments in detail and provide a revised manuscript using ‘track changes’ to indicate where the paper has changed.

Reviewer #1

1. Major: The problem analysis does not fit the research question. Bade on the introduction, the reader expects the paper to focus on the complex relationship between PS and outcomes and especially the underlying mechanisms. Instead the paper focuses on the evidence of the relationship and its implications for interventions.

Our Response: We have rewritten the introduction section to better reflect the complex relationship between PS and various outcomes.

2. Following the introduction, the research question should be wider and especially include the subjective patient safety outcomes in order to unravel this complex relationship. OR the introduction should be complete rewritten in order to focus on why we need a summary of evidence for this specific relationship, why is it so important to show the reader this overview of evidence. In addition, also argue why this overview does not include the subjective safety outcomes.

Our response: Previous systematic reviews on psychological safety in healthcare (i.e., Grailey et al, 2021; O’Donovan & McAuliffe, 2020) have reviewed papers that reported on the relationship between psychological safety and self-reported outcomes. However, it is widely acknowledged that self-report survey measures are limited by self-report bias and response fatigue (Donaldson & Grant-Vallone, 2002: Newman et al, 2017). Moreover, there is evidence that self-reports can artificially inflate the relationship between safety climate and safety outcomes (Beus et al., 2010), whereas evaluations of hospital patient safety performance (as well as fines, suspensions and closures) are based on objectively reported metrics (e.g., medical errors, critical incidents). Our aim in this paper was to provide a new contribution to the field in line with the current literature highlighting the complexity of defining patient safety and the need for updated approaches to patient safety definitions. Thus, we focused on studies that reported the relationship between PS and objective/robust measures of patient safety. Recent reviews of the PS area report on the positive significant relationships between high levels of psychological safety with learning and creativity outcomes (e.g., Edmondson & Bransby, 2023). However, this trend is not observed in healthcare (we cite the relevant research in our introduction). Therefore, we believe that a review of the evidence on the relationship between psychological safety and objective measures of patient safety would provide a new and needed addition to the literature. We hope that the reviewer agrees with our sentiments. We would also like to add that our collective experiences of teaching healthcare professionals indicate a desire from healthcare workers to be informed of the evidence concerning the impact of psychological safety on patient safety ‘in practice’.

References

Beus JM, Payne SC, Bergman ME, Arthur Jr W. Safety climate and injuries: an examination of theoretical and empirical relationships. Journal of applied psychology. 2010 Jul;95(4):713.

Donaldson SI, Grant-Vallone EJ. Understanding self-report bias in organizational behavior research. J Bus Psychol 2002;17:245–60.

Edmondson AC, Bransby DP. Psychological safety comes of age: observed themes in an established literature. Annual Review of Organizational Psychology and Organizational Behavior. 2023 Feb 2;10(1).

Grailey KE, Murray E, Reader T, Brett SJ. The presence and potential impact of psychological safety in the healthcare setting: an evidence synthesis. BMC health services research. 2021 Dec;21:1-5.

Newman A, Donohue R, Eva N. Psychological safety: a systematic review of the literature. Hum Resour Manag Rev 2017;27:521–35.

O’donovan R, Mcauliffe E. A systematic review of factors that enable psychological safety in healthcare teams. International journal for quality in health care. 2020 May;32(4):240-50.

3. Minor: Why is there a heading called “psychological safety and psychological safety”

Our response: Thank you for pointing this out. Apologies, we have corrected this typo.

4. Methods: In the future you might consider using tools to assist you in a systematic review, such as Ryyan or ASreview

Our response: We agree with the point of the reviewer. However, please note that we did use Rayyan in our review process. See - “Eligibility Criteria and Study Selection,” “Duplicate control and title and abstract review were conducted using Rayyan”.

5. You might consider identifying other reviews on PS. What did and didn't they conclude. Did they also include your selection of papers. Have you identified new papers or was this review also be able to be conducted based on the present reviews?

Our response: We have cited and included a large number of reviews on PS in our paper (see list below). The papers were useful in identifying the gaps in the literature. We highlight in the introduction the added value of our paper – in that it addresses an area that has not been previously covered in the literature. As we have previously mentioned in point 2, review papers to date have highlighted the problem surrounding the overreliance on self-report measures in terms of common method variance.

Here is a list of the included review papers in our manuscript:

Edmondson AC, Lei Z. Psychological safety: The history, renaissance, and future of an interpersonal construct. Annual Review of Organizational Psychology and Organizational Behavior. 2014 Mar 21;1(1):23–43.

Edmondson AC, Bransby DP. Psychological safety comes of age: observed themes in an established literature. Annual Review of Organizational Psychology and Organizational Behavior. 2023 Feb 2;10(1).

Frazier ML, Fainshmidt S, Klinger RL, Pezeshkan A, Vracheva V. Psychological safety: A meta-analytic review and extension. Personnel Psychology. 2017 Oct 14;70(1):113–65.

Grailey KE, Murray E, Reader T, Brett SJ. The presence and potential impact of psychological safety in the healthcare setting: an evidence synthesis. BMC health services research [Internet]. 2021 Aug 5 [cited 2021 Nov 4];21(1):773.

Liu JW, Ein N, Plouffe RA, Gervasio J, St. Cyr K, Nazarov A, Richardson JD. Meta-Analysis and Systematic Review of the Measures of Psychological Safety. medRxiv. 2024:2024-02.

O’donovan R, Mcauliffe E. A systematic review of factors that enable psychological safety in healthcare teams. International Journal for Quality in Health Care. 2020 Mar 31;32(4):240–50.

6. Results: - In the interpretation of the relationship there is a lack of nuanced. If a relationship between PS and medical errors is found, this means that people are more or less willing to report errors, it is not the same as the actual errors made. One could argue both ways: high PS could make the reporting of errors less urging and therefore lower OR high PS could make the reporting of errors more natural in an open and learning environment. This is somewhat later on discussed but not incorporated in the presentation of the results at the beginning of the Result section.

Our response: We thank the reviewer for the opportunity to improve the quality of the Results section. We have now addressed the difficulties of a “straightforward interpretation” at the beginning of the Result section entitled - (“Is there a relationship between Psychological Safety and Patient Safety?) section”.

We have included the following text: “PS and positive objective patient safety outcomes are sometimes “difficult” to detect and resistant to linear interpretation, as PS seems to share contextual dependencies between objective reporting behavior and patient safety. Additionally, high reporting rates may indicate high PS, where incidents are openly discussed and used as learning opportunities and protocol compliance. Conversely, low reporting rates might be interpreted as evidence of enhanced team performance, although this could also reflect underreporting or fear of disclosure.”

7. There is a lack of insights presented in teamwork in healthcare. People that filled in the questionnaire of Edmondons are healthcare professionals. They are part of multiple teams; multi teammembership. What are the teams we are referring to in the result section. Are that the multidisciplinairy teams that provide care? Or the monodisciplinairy teams? That is a big difference in relation to PS. In multidisciplinairy teams the different disciplines are stables but not the people representing this discipline.

Our response: We agree with the point of the reviewer. We do mention this as one of the limitations of the Edmondson questionnaire (see our section on Methodology and Sampling Issues), in that it's not possible to know what ‘team’ respondents have in mind when they fill in the questionnaire. The teamwork element is an important factor in this story, but our review of the nine papers did not allow us to reach any substantive conclusions on this issue.

8. Discussion/conclusion: What is the answer to the second part of the research question? Not clear what this review adds. The paradoxes where already know prior to the research. Make clear what we did not know and after the review do know. After rewriting the introduction, have in mind that the introduction and discussion should cover other angles. Now they are too similar in their message.

Our response: We hope that our rewriting of the paper has illuminated more clearly what our review has contributed to the field. We agree with the reviewer that the paradoxes were already mentioned in the literature, however – we would argue that our paper is the first attempt to review the paradox between PS and patient safety in detail. Beyond this, we can provide a clearer description of what our paper contributes:

What is already known on this topic – High levels of psychological safety in healthcare teams has been advocated as an important mechanism by which patient safety can be improved. However, the majority of the research is based on self-report measures.

What this study adds – This is the first systematic review to assess the relationship between psychological safety and objectively measured patient safety. It identifies that commonly used definitions of patient safety as simply the absence of harm, might need to broaden to concepts of contemporary safety science.

How this study might affect research, practice or policy – High levels of psychological safety has been advocated as an important mechanism by which patient safety can be improved, but there is insufficient evidence to support this idea due to the framing of safety only as the absence of harm in the healthcare literature, rather than a safety science lens

Reviewer #2:

1. This is a really interesting and important paper and congratulations to the authors. The comments made here are in the spirit of strengthening the paper.

Our response: We appreciate the reviewer’s perspective, and we thank for the positive appraisal of the revised manuscript, as well as for the opportunity to further enhance the quality of the paper.

2. The definition of psychological safety is important. Edmondson’s 1999 definition and measurement is not just about me being able to take risks (express vulnerability) in relation to what I do and do not know but also relates to me being able to question my colleagues on their actions and how correct/safe they are. E.g. it is an important point made in the introduction that in healthcare a lot of time is spent trying to figure out what is safe to talk about – but is this not what psychological safety is about?

Our response: We agree with the point of the reviewer that the Edmondson approach is about taking risks but also relates to speaking up to colleagues. We acknowledge this problem in terms of our discussion of the inherent challenges in using the Edmondson questionnaire and what it means for the validity of psychological safety. For example, we mention the problem of what individuals have in mind when thinking about what ‘team’ the questionnaire refers to. Additionally, we note the reliability and unidimensional limitations of the Edmondson questionnaire. The point about speaking up is an important one, but the available research evidence in organizational behaviour literature tends to treat employee silence and employee voice as distinct constructs from psychological safety. However, this is a bigger problem concerning silence and voice in healthcare per se, with a recent integrative review indicating a need for further research regarding the distinction between what drives safety voice versus general employee voice, and how both voice and silence can operate in parallel in healthcare (Lainidi et al, 2023). We agree with the reviewer that a lack of psychological safety means either uncertainty on whether doing the right thing will be accepted or even certainty that other issues (e.g., the reputation of the unit/hospital) should be prioritised over doing the right thing by e.g., the patients. In that sense, speaking up can be included under the “taking a risk” umbrella. In practical terms, healthcare professionals might experience the high levels psychological safety in their team/unit, while also knowing there are certain topics that should not be voiced.

Lainidi O, Jendeby MK, Montgomery A, Mouratidis C, Paitaridou K, Cook C, Johnson J, Karakasidou E. An integrative systematic review of employee silence and voice in healthcare: what are we really measuring?. Frontiers in Psychiatry. 2023 May 25;14:1111579.

3. There are more nuanced definitions of patient safety other than freedom from harm and as you argue this is not a good definition. I think these other definitions informed by Safety Management Systems work in aviation and other fields should also be included here?

Our response: We thank the reviewer for this suggestion. We have now extended the definition to state: “Defining the commitment to advancing safe care, patient safety—the most enduring and foundational principle of medicine—represents the core value of healthcare quality by emphasizing freedom from any harm associated with health care in clinical practices.”

4. Section – Psychological safety and should be ‘patient’ safety?

Our response: Thank you for pointing this out. Apologies, we have corrected this typo.

5. Line 80 - Medical errors third leading cause of death – needs to be nuanced. https://qualitysafety.bmj.com/content/26/5/423

Our response: We thank the reviewer for this advice. We have now corrected the percentage to “Medical errors are translating into over three million deaths globally each year” as reported in Global patient safety report 2024. Geneva: World Health Organization; 2024.

6. Lines 85-90 Should psychological safety not also lead to increased reporting of adverse events and to increased challenges to each other on unsafe practices?

Our response: We thank the reviewer for raising this crucial point. In healthcare, where professionals are highly trained and educated, psychological safety should lead to increased reporting of adverse events/incidents/errors and more frequent challenges to unsafe practices only when healthcare professionals perceive reporting as a constructive action aimed at improving patient safety and delivering high-quality care, rather than as an admission of personal failure or accountability. However, this association seems to be quite complex as psychological safety could also lead to decreased reporting of adverse events/incidents/errors, while also there is not concrete evidence on “how much” psychological safety is needed or whether too much psychological safety can lead to unnecessary risks or aversion towards formal reporting. We agree with this point and it links to the reviewer’s earlier comment on what psychological safety is – and this point indirectly implies that academics and practitioners might need to think what is NOT psychological safety. To address this, in the Results section (at the beginning of - Is there a relationship b

---

## [Decision Letter · Decision Letter 1]

16 Feb 2025

PONE-D-24-42345R1Psychological safety and patient safety: a systematic and narrative reviewPLOS ONE

Dear Dr. Montgomery,

Thank you for submitting your manuscript to PLOS ONE. After careful consideration, we feel that it has merit but does not fully meet PLOS ONE’s publication criteria as it currently stands. Therefore, we invite you to submit a revised version of the manuscript that addresses the points raised during the review process.

We look forward to receiving your revised manuscript.

Kind regards,

Wen Wu, phd

Academic Editor

PLOS ONE

Additional Editor Comments:

Dear Authors,

Thanks for your effort in revising the last version of the manuscript. I have carefully read the whole submission. Generally, I agree with the feedback from three reviewers. Their comments are varied, ranging from major and minor revisions. I give you the opportunity to revise and resubmit the manuscript. They pointed out that this research project is interesting and significant (R 2 &3). However, they mentioned lots of weaknesses you have to solve. For instance, the introduction should be improved. And the significance of your research in healthcare context should be further shown. I hope you can respond to their comments one by one, and make detailed improvement in the manuscript in the next round.

Reviewer 1:

Introduction:

- Major: The problem analysis does not fit the research question. Bade on the introduction, the reader expects the paper to focus on the complex relationship between PS and outcomes and especially the underlying mechanisms. Instead the paper focuses on the evidence of the relationship and its implications for interventions. Following the introduction, the research question should be wider and especially include the subjective patient safety outcomes in order to unravel this complex relationship. OR the introduction should be complete rewritten in order to focus on why we need a summary of evidence for this specific relationship, why is it so important to show the reader this overview of evidence. In addition, also argue why this overview does not include the subjective safety outcomes.

- Minor: Why is there a heading called “psychological safety and psychological safety”

Methods:

- In the future you might consider using tools to assist you in a systematic review, such as Ryyan or ASreview

- You might consider identifying other reviews on PS. What did and didn't they conclude. Did they also include your selection of papers. Have you identified new papers or was this review also be able to be conducted based on the present reviews?

Results:

- In the interpretation of the relationship there is a lack of nuanced. If a relationship between PS and medical errors is found, this means that people are more or less willing to report errors, it is not the same as the actual errors made. One could argue both ways: high PS could make the reporting of errors less urging and therefore lower OR high PS could make the reporting of errors more natural in an open and learning environment. This is somewhat later on discussed but not incorporated in the presentation of the results at the beginning of the Result section.

- There is a lack of insights presented in teamwork in healthcare. People that filled in the questionnaire of Edmondons are healthcare professionals. They are part of multiple teams; multi teammembership. What are the teams we are referring to in the result section. Are that the multidisciplinairy teams that provide care? Or the monodisciplinairy teams? That is a big difference in relation to PS. In multidisciplinairy teams the different disciplines are stables but not the people representing this discipline.

Discussion/conclusion:

- What is the answer to the second part of the research question?

- Not clear what this review adds. The paradoxes where already know prior to the research. Make clear what we did not know and after the review do know.

- After rewriting the introduction, have in mind that the introduction and discussion should cover other angles. Now they are too similar in their message.

Reviewer 2:

This is a really interesting and important paper and congratulations to the authors. The comments made here are in the spirit of strengthening the paper.

The definition of psychological safety is important. Edmondson’s 1999 definition and measurement is not just about me being able to take risks (express vulnerability) in relation to what I do and do not know but also relates to me being able to question my colleagues on their actions and how correct/safe they are. E.g. it is an important point made in the introduction that in healthcare a lot of time is spent trying to figure out what is safe to talk about – but is this not what psychological safety is about?

There are more nuanced definitions of patient safety other than freedom from harm and as you argue this is not a good definition. I think these other definitions informed by Safety Management Systems work in aviation and other fields should also be included here?

Section – Psychological safety and should be ‘patient’ safety?

Line 80 - Medical errors third leading cause of death – needs to be nuanced

https://qualitysafety.bmj.com/content/26/5/423

Lines 85-90 Should psychological safety not also lead to increased reporting of adverse events and to increased challenges to each other on unsafe practices?

Methods

Search strategy

Line 112 – why were only quantitative studies included – needs justification

Results

Line 138 not sure why female participants pulled out in particular – would it be more appropriate to say ‘Only 4 out of 9 studies noted gender of participants’.

Line 142 needs more discussion on what tools other than Edmondson were used.

Discussion

Line 279 – or improved objective patient safety measures (rather than just formal reporting or errors?) and what are these?

Reviewer 3:

This an important systematic review of the evidence linking psychological safety and patient safety. The results will be of great interest to specialists in the patient safety domain. I think the manuscript could be greatly improved if the authors were to present their review from a different stance. Rather than presenting a list of the findings of the nine studies, I suggest a more nuanced approach which recognises the complexity of the link between the two entities. We know already that healthcare teams with high psych safety can appear to be operating less safely because they tend to report more patient safety incidents and unsafe practices. This could be the main focus of the research rather than a straight forward yes/no research question. The sections towards the end of the manuscript nicely address this and they could be moved to the beginning of the manuscript as the background to the review.

I think the results section should tell a 'story' rather than a 'a laundry list' of study findings. The study findings could be used as evidence for the 'story'. This would address an issue in the content where the authors report the study findings and then make a conclusion. It would be more interesting to reverse this process.

In my view (a) and (b) in the research question are not necessary and take from the impact of the question itself.

I also have some typos and writing style issues spotted which I hope are helpful:

The 0-10 rule is inconsistently applied. i.e. numbers between 0-10 should be written in long hand (abstract and L139, 8/9 7/9 etc )

L57, Should 'of' be deleted ?

Ideally sentences that begin with XXXXX found/demonstrated ........are not ideal....Instead state the finding and include the author reference at the end where possible

L 151 inconsistent use of capital letters

L92 rewrite' Therefore the direction or existence of causality between patient and PS is not clear'

Psychological safety is shortened early on to PS but this is not then used consistently ie Lines 96,207,209,214,228,255,256,259,262,264,268,270,273,274,277,282,296,304,307,318,325,330,331,334,337,336,337,338. Personally I prefer the phrase rather than the acronym as its easier for the reader.

L 225 paper or papers ?

I would love to see this manuscript published as it would add value to the body of knowledge if it was to be rewritten more creatively.

Reviewers' comments:

Reviewer's Responses to Questions

**Comments to the Author**

1. If the authors have adequately addressed your comments raised in a previous round of review and you feel that this manuscript is now acceptable for publication, you may indicate that here to bypass the “Comments to the Author” section, enter your conflict of interest statement in the “Confidential to Editor” section, and submit your "Accept" recommendation.

Reviewer #2: All comments have been addressed

Reviewer #3: (No Response)

2. Is the manuscript technically sound, and do the data support the conclusions?

Reviewer #2: (No Response)

Reviewer #3: Yes

3. Has the statistical analysis been performed appropriately and rigorously?

Reviewer #2: (No Response)

Reviewer #3: N/A

4. Have the authors made all data underlying the findings in their manuscript fully available?

Reviewer #2: (No Response)

Reviewer #3: Yes

5. Is the manuscript presented in an intelligible fashion and written in standard English?

Reviewer #2: (No Response)

Reviewer #3: No

6. Review Comments to the Author

Reviewer #2: (No Response)

Reviewer #3: Thankyou for re-arranging the text of the manuscript and for your responses to the points made. Unfortunately I can see some pronouns are still missing (eg L 40 in the abstract 'are' is missing before the 'from the USA.) Psychological safety is writtenin full on L 84.'a' is missing on L86. there are others too numerous to list.In addition both american and English spelling is used. eg organizational vs characterised. There are commas missing in lots of sentences and many of the sentences in the introduction and elsewhere are too long and need to be broken up so that you bring the reader with you in your statements.

7. PLOS authors have the option to publish the peer review history of their article (what does this mean? ). If published, this will include your full peer review and any attached files.

**Do you want your identity to be public for this peer review?** For information about this choice, including consent withdrawal, please see our Privacy Policy .

Reviewer #2: No

Reviewer #3: No

---

## [Author Response · Author response to Decision Letter 2]

17 Feb 2025

Dear Dr. Wan and Reviewers,

We want to thank you and the three reviewers for their constructive feedback on our paper. Their suggestions improved the paper. In the following letter, we address all three reviewers' comments in detail and provide a revised manuscript using ‘track changes’ to indicate where the paper has changed.

Reviewer #1

1. Major: The problem analysis does not fit the research question. Bade on the introduction, the reader expects the paper to focus on the complex relationship between PS and outcomes and especially the underlying mechanisms. Instead the paper focuses on the evidence of the relationship and its implications for interventions.

Our Response: We have rewritten the introduction section to better reflect the complex relationship between PS and various outcomes.

2. Following the introduction, the research question should be wider and especially include the subjective patient safety outcomes in order to unravel this complex relationship. OR the introduction should be complete rewritten in order to focus on why we need a summary of evidence for this specific relationship, why is it so important to show the reader this overview of evidence. In addition, also argue why this overview does not include the subjective safety outcomes.

Our response: Previous systematic reviews on psychological safety in healthcare (i.e., Grailey et al, 2021; O’Donovan & McAuliffe, 2020) have reviewed papers that reported on the relationship between psychological safety and self-reported outcomes. However, it is widely acknowledged that self-report survey measures are limited by self-report bias and response fatigue (Donaldson & Grant-Vallone, 2002: Newman et al, 2017). Moreover, there is evidence that self-reports can artificially inflate the relationship between safety climate and safety outcomes (Beus et al., 2010), whereas evaluations of hospital patient safety performance (as well as fines, suspensions and closures) are based on objectively reported metrics (e.g., medical errors, critical incidents). Our aim in this paper was to provide a new contribution to the field in line with the current literature highlighting the complexity of defining patient safety and the need for updated approaches to patient safety definitions. Thus, we focused on studies that reported the relationship between PS and objective/robust measures of patient safety. Recent reviews of the PS area report on the positive significant relationships between high levels of psychological safety with learning and creativity outcomes (e.g., Edmondson & Bransby, 2023). However, this trend is not observed in healthcare (we cite the relevant research in our introduction). Therefore, we believe that a review of the evidence on the relationship between psychological safety and objective measures of patient safety would provide a new and needed addition to the literature. We hope that the reviewer agrees with our sentiments. We would also like to add that our collective experiences of teaching healthcare professionals indicate a desire from healthcare workers to be informed of the evidence concerning the impact of psychological safety on patient safety ‘in practice’.

References

Beus JM, Payne SC, Bergman ME, Arthur Jr W. Safety climate and injuries: an examination of theoretical and empirical relationships. Journal of applied psychology. 2010 Jul;95(4):713.

Donaldson SI, Grant-Vallone EJ. Understanding self-report bias in organizational behavior research. J Bus Psychol 2002;17:245–60.

Edmondson AC, Bransby DP. Psychological safety comes of age: observed themes in an established literature. Annual Review of Organizational Psychology and Organizational Behavior. 2023 Feb 2;10(1).

Grailey KE, Murray E, Reader T, Brett SJ. The presence and potential impact of psychological safety in the healthcare setting: an evidence synthesis. BMC health services research. 2021 Dec;21:1-5.

Newman A, Donohue R, Eva N. Psychological safety: a systematic review of the literature. Hum Resour Manag Rev 2017;27:521–35.

O’donovan R, Mcauliffe E. A systematic review of factors that enable psychological safety in healthcare teams. International journal for quality in health care. 2020 May;32(4):240-50.

3. Minor: Why is there a heading called “psychological safety and psychological safety”

Our response: Thank you for pointing this out. Apologies, we have corrected this typo.

4. Methods: In the future you might consider using tools to assist you in a systematic review, such as Ryyan or ASreview

Our response: We agree with the point of the reviewer. However, please note that we did use Rayyan in our review process. See - “Eligibility Criteria and Study Selection,” “Duplicate control and title and abstract review were conducted using Rayyan”.

5. You might consider identifying other reviews on PS. What did and didn't they conclude. Did they also include your selection of papers. Have you identified new papers or was this review also be able to be conducted based on the present reviews?

Our response: We have cited and included a large number of reviews on PS in our paper (see list below). The papers were useful in identifying the gaps in the literature. We highlight in the introduction the added value of our paper – in that it addresses an area that has not been previously covered in the literature. As we have previously mentioned in point 2, review papers to date have highlighted the problem surrounding the overreliance on self-report measures in terms of common method variance.

Here is a list of the included review papers in our manuscript:

Edmondson AC, Lei Z. Psychological safety: The history, renaissance, and future of an interpersonal construct. Annual Review of Organizational Psychology and Organizational Behavior. 2014 Mar 21;1(1):23–43.

Edmondson AC, Bransby DP. Psychological safety comes of age: observed themes in an established literature. Annual Review of Organizational Psychology and Organizational Behavior. 2023 Feb 2;10(1).

Frazier ML, Fainshmidt S, Klinger RL, Pezeshkan A, Vracheva V. Psychological safety: A meta-analytic review and extension. Personnel Psychology. 2017 Oct 14;70(1):113–65.

Grailey KE, Murray E, Reader T, Brett SJ. The presence and potential impact of psychological safety in the healthcare setting: an evidence synthesis. BMC health services research [Internet]. 2021 Aug 5 [cited 2021 Nov 4];21(1):773.

Liu JW, Ein N, Plouffe RA, Gervasio J, St. Cyr K, Nazarov A, Richardson JD. Meta-Analysis and Systematic Review of the Measures of Psychological Safety. medRxiv. 2024:2024-02.

O’donovan R, Mcauliffe E. A systematic review of factors that enable psychological safety in healthcare teams. International Journal for Quality in Health Care. 2020 Mar 31;32(4):240–50.

6. Results: - In the interpretation of the relationship there is a lack of nuanced. If a relationship between PS and medical errors is found, this means that people are more or less willing to report errors, it is not the same as the actual errors made. One could argue both ways: high PS could make the reporting of errors less urging and therefore lower OR high PS could make the reporting of errors more natural in an open and learning environment. This is somewhat later on discussed but not incorporated in the presentation of the results at the beginning of the Result section.

Our response: We thank the reviewer for the opportunity to improve the quality of the Results section. We have now addressed the difficulties of a “straightforward interpretation” at the beginning of the Result section entitled - (“Is there a relationship between Psychological Safety and Patient Safety?) section”.

We have included the following text: “PS and positive objective patient safety outcomes are sometimes “difficult” to detect and resistant to linear interpretation, as PS seems to share contextual dependencies between objective reporting behavior and patient safety. Additionally, high reporting rates may indicate high PS, where incidents are openly discussed and used as learning opportunities and protocol compliance. Conversely, low reporting rates might be interpreted as evidence of enhanced team performance, although this could also reflect underreporting or fear of disclosure.”

7. There is a lack of insights presented in teamwork in healthcare. People that filled in the questionnaire of Edmondons are healthcare professionals. They are part of multiple teams; multi teammembership. What are the teams we are referring to in the result section. Are that the multidisciplinairy teams that provide care? Or the monodisciplinairy teams? That is a big difference in relation to PS. In multidisciplinairy teams the different disciplines are stables but not the people representing this discipline.

Our response: We agree with the point of the reviewer. We do mention this as one of the limitations of the Edmondson questionnaire (see our section on Methodology and Sampling Issues), in that it's not possible to know what ‘team’ respondents have in mind when they fill in the questionnaire. The teamwork element is an important factor in this story, but our review of the nine papers did not allow us to reach any substantive conclusions on this issue.

8. Discussion/conclusion: What is the answer to the second part of the research question? Not clear what this review adds. The paradoxes where already know prior to the research. Make clear what we did not know and after the review do know. After rewriting the introduction, have in mind that the introduction and discussion should cover other angles. Now they are too similar in their message.

Our response: We hope that our rewriting of the paper has illuminated more clearly what our review has contributed to the field. We agree with the reviewer that the paradoxes were already mentioned in the literature, however – we would argue that our paper is the first attempt to review the paradox between PS and patient safety in detail. Beyond this, we can provide a clearer description of what our paper contributes:

What is already known on this topic – High levels of psychological safety in healthcare teams has been advocated as an important mechanism by which patient safety can be improved. However, the majority of the research is based on self-report measures.

What this study adds – This is the first systematic review to assess the relationship between psychological safety and objectively measured patient safety. It identifies that commonly used definitions of patient safety as simply the absence of harm, might need to broaden to concepts of contemporary safety science.

How this study might affect research, practice or policy – High levels of psychological safety has been advocated as an important mechanism by which patient safety can be improved, but there is insufficient evidence to support this idea due to the framing of safety only as the absence of harm in the healthcare literature, rather than a safety science lens

Reviewer #2:

1. This is a really interesting and important paper and congratulations to the authors. The comments made here are in the spirit of strengthening the paper.

Our response: We appreciate the reviewer’s perspective, and we thank for the positive appraisal of the revised manuscript, as well as for the opportunity to further enhance the quality of the paper.

2. The definition of psychological safety is important. Edmondson’s 1999 definition and measurement is not just about me being able to take risks (express vulnerability) in relation to what I do and do not know but also relates to me being able to question my colleagues on their actions and how correct/safe they are. E.g. it is an important point made in the introduction that in healthcare a lot of time is spent trying to figure out what is safe to talk about – but is this not what psychological safety is about?

Our response: We agree with the point of the reviewer that the Edmondson approach is about taking risks but also relates to speaking up to colleagues. We acknowledge this problem in terms of our discussion of the inherent challenges in using the Edmondson questionnaire and what it means for the validity of psychological safety. For example, we mention the problem of what individuals have in mind when thinking about what ‘team’ the questionnaire refers to. Additionally, we note the reliability and unidimensional limitations of the Edmondson questionnaire. The point about speaking up is an important one, but the available research evidence in organizational behaviour literature tends to treat employee silence and employee voice as distinct constructs from psychological safety. However, this is a bigger problem concerning silence and voice in healthcare per se, with a recent integrative review indicating a need for further research regarding the distinction between what drives safety voice versus general employee voice, and how both voice and silence can operate in parallel in healthcare (Lainidi et al, 2023). We agree with the reviewer that a lack of psychological safety means either uncertainty on whether doing the right thing will be accepted or even certainty that other issues (e.g., the reputation of the unit/hospital) should be prioritised over doing the right thing by e.g., the patients. In that sense, speaking up can be included under the “taking a risk” umbrella. In practical terms, healthcare professionals might experience the high levels psychological safety in their team/unit, while also knowing there are certain topics that should not be voiced.

Lainidi O, Jendeby MK, Montgomery A, Mouratidis C, Paitaridou K, Cook C, Johnson J, Karakasidou E. An integrative systematic review of employee silence and voice in healthcare: what are we really measuring?. Frontiers in Psychiatry. 2023 May 25;14:1111579.

3. There are more nuanced definitions of patient safety other than freedom from harm and as you argue this is not a good definition. I think these other definitions informed by Safety Management Systems work in aviation and other fields should also be included here?

Our response: We thank the reviewer for this suggestion. We have now extended the definition to state: “Defining the commitment to advancing safe care, patient safety—the most enduring and foundational principle of medicine—represents the core value of healthcare quality by emphasizing freedom from any harm associated with health care in clinical practices.”

4. Section – Psychological safety and should be ‘patient’ safety?

Our response: Thank you for pointing this out. Apologies, we have corrected this typo.

5. Line 80 - Medical errors third leading cause of death – needs to be nuanced. https://qualitysafety.bmj.com/content/26/5/423

Our response: We thank the reviewer for this advice. We have now corrected the percentage to “Medical errors are translating into over three million deaths globally each year” as reported in Global patient safety report 2024. Geneva: World Health Organization; 2024.

6. Lines 85-90 Should psychological safety not also lead to increased reporting of adverse events and to increased challenges to each other on unsafe practices?

Our response: We thank the reviewer for raising this crucial point. In healthcare, where professionals are highly trained and educated, psychological safety should lead to increased reporting of adverse events/incidents/errors and more frequent challenges to unsafe practices only when healthcare professionals perceive reporting as a constructive action aimed at improving patient safety and delivering high-quality care, rather than as an admission of personal failure or accountability. However, this association seems to be quite complex as psychological safety could also lead to decreased reporting of adverse events/incidents/errors, while also there is not concrete evidence on “how much” psychological safety is needed or whether too much psychological safety can lead to unnecessary risks or aversion towards formal reporting. We agree with this point and it links to the reviewer’s earlier comment on what psychological safety is – and this point indirectly implies that academics and practitioners might need to think what is NOT psychological safety. To address this, in the Results section (at the beginning of - Is there a relationship between Psychological Safety an

---

## [Decision Letter · Decision Letter 2]

18 Mar 2025

Psychological safety and patient safety: a systematic and narrative review

PONE-D-24-42345R2

Dear Dr. Montgomery,

We’re pleased to inform you that your manuscript has been judged scientifically suitable for publication and will be formally accepted for publication once it meets all outstanding technical requirements.

Kind regards,

Wen Wu, phd

Academic Editor

PLOS ONE

Additional Editor Comments (optional):

Dear authors,

Thanks for your effort in revising the submission. Your revised manuscript had been reviewed by three experts in this field. I also have carefully read the manuscript.

Generally I can see the effort in improving the manuscript. Based on comments from reviewers and my reading, I trust this research project can make good contributions to literature and I am pleased to offer acceptance of this paper.

Thanks for all your effort and your interest in PLOS One. Hope you submit more works to this journal!

Best regards,

Reviewers' comments:

Reviewer's Responses to Questions

**Comments to the Author**

1. If the authors have adequately addressed your comments raised in a previous round of review and you feel that this manuscript is now acceptable for publication, you may indicate that here to bypass the “Comments to the Author” section, enter your conflict of interest statement in the “Confidential to Editor” section, and submit your "Accept" recommendation.

Reviewer #1: All comments have been addressed

Reviewer #2: All comments have been addressed

Reviewer #3: All comments have been addressed

2. Is the manuscript technically sound, and do the data support the conclusions?

Reviewer #1: Yes

Reviewer #2: (No Response)

Reviewer #3: Yes

3. Has the statistical analysis been performed appropriately and rigorously?

Reviewer #1: N/A

Reviewer #2: (No Response)

Reviewer #3: N/A

4. Have the authors made all data underlying the findings in their manuscript fully available?

Reviewer #1: (No Response)

Reviewer #2: (No Response)

Reviewer #3: Yes

5. Is the manuscript presented in an intelligible fashion and written in standard English?

Reviewer #1: Yes

Reviewer #2: (No Response)

Reviewer #3: Yes

6. Review Comments to the Author

Reviewer #1: Thank you for addressing all the comments. You have done a great job in commenting all comments and adjusting the manuscript accordingly.

Reviewer #2: (No Response)

Reviewer #3: There are still a number of grammatical issues and spelling inconsistencies which at the third phase of the review process should have been corrected. In the abstract, 8 databases should be eight databases.

L65 'not cover up' should be replaced with 'report/declare/expose. There is a full stop missing after [9,10]. This should be followed by a new sentence as the sentence is too long.

American and English spelling is still in evidence

L66 organisational

L35 analyzed

L79 behaviours vs L53 behaviors

L84 psychological safety rather than PS

L116 emphasizing

L118 recognising

L119 full stop missing after [29]

L145 summarize

7. PLOS authors have the option to publish the peer review history of their article (what does this mean? ). If published, this will include your full peer review and any attached files.

**Do you want your identity to be public for this peer review?** For information about this choice, including consent withdrawal, please see our Privacy Policy .

Reviewer #1: No

Reviewer #2: No

Reviewer #3: No

---

## [Editor Report · Acceptance letter]

PONE-D-24-42345R2

PLOS ONE

Dear Dr. Montgomery,

I'm pleased to inform you that your manuscript has been deemed suitable for publication in PLOS ONE. Congratulations! Your manuscript is now being handed over to our production team.

Kind regards,

on behalf of

Dr. Wen Wu

Academic Editor

PLOS ONE